# Evaluation of the effect of formalin fixation on skin specimens in dogs and cats

Jaimie L. Miller[1] and Michael J. Dark[2,3]

[1] College of Veterinary Medicine, University of Florida, Gainesville, FL, USA
[2] Department of Infectious Diseases and Pathology, College of Veterinary Medicine, University of Florida, Gainesville, FL, USA
[3] Emerging Pathogens Institute, University of Florida, Gainesville, FL, USA

## ABSTRACT

Skin and subcutaneous tissues are the origin of most common neoplasms affecting dogs, accounting for approximately one third of all tumors encountered in the species. Surgical excision is frequently the best chance for a cure; determining factors influencing the success of excision are vital for surgical management of cases. This work examined the shrinkage of skin of various lengths from three sites in formalin for both dogs and cats. Tissues were measured on the animal (initial measurement), at the time of excision (post-removal), and after formalin fixation (post-fixation). While shrinkage after tissue removal was found in samples from the thorax, abdomen, and rear leg in dogs and from the rear leg in cats, no significant shrinkage due to formalin fixation was detected in any sample except for the thoracic samples from the dog. Therefore, when determining where to make incisions to effect a surgical cure, initial measurements should take into account tissue shrinkage effects.

## INTRODUCTION

Skin and subcutaneous tissues are common sites of neoplasms affecting animals. These are the most common neoplasms of dogs, accounting for approximately one third of all tumors encountered in the species. In the cat, skin and subcutaneous neoplasms are second only to lymphoid neoplasms and account for approximately one fourth of all tumors in the species (*Vail & Withrow, 2007*).

A recurring theme in surgical management of cancer in both species is that the first surgery has the best chance of cure (*Vail & Withrow, 2007*); therefore, it is imperative to obtain sufficient normal tissue to ensure all neoplastic tissue has been removed.

There are two important considerations for margins with neoplasms; first, the gross margin determined by the surgeon, and second, the margin assessed histologically by the pathologist. A variety of recommendations as to sufficient margins for surgical cure have been issued for different neoplasms. For example, 2 cm surgical margins are recommended for the surgical cure of grade I and II mast cell tumors (*Fulcher et al., 2006*; *Simpson et al., 2004*). Histologic evaluation of surgical margins has been used to determine the likelihood of recurrence for soft tissue sarcomas and carcinomas (*Scarpa et al., 2012*). If the tissue

Corresponding author
Michael J. Dark, darkmich@ufl.edu

dimensions *in vivo* and post-excision differ significantly, assessment of tumor margin clearance from fixed specimens will be misleading, particularly if there are differences in the normal and tumor tissue responses to formaldehyde fixation (*Hudson-Peacock, Matthews & Lawrence, 1995*).

While there have been previous studies examining tissue shrinkage in formalin in people and animals (*Blasdale et al., 2010*; *Dauendorffer et al., 2009*; *Kerns et al., 2008*; *Reimer et al., 2005*), none have compared the effects on multiple species. To assess this, we performed a pilot study using post-mortem skin samples from ten cats and ten dogs to compare the degree of shrinkage via fixation in 10% neutral buffered formalin.

## MATERIALS AND METHODS

This study was approved by the University of Florida Institutional Animal Care and Use Committee (approval #200902876). Clinically normal adult dogs (10) and cats (10) collected from a local animal shelter were used in this study, after being humanely euthanized for unrelated reasons. Five males and five females were randomly selected for each group based on being grossly free of any skin defects. Skin samples were collected within two hours of euthanasia. All animals were randomly placed in left or right lateral recumbency and the hair clipped for collection of skin from the thorax, abdomen, and distal lateral hind limb. A calibrated digital carbon fiber caliper (15-077-957; Fisher Scientific, Pittsburg, PA) was used to measure 1.0 cm, 2.0 cm, 3.0 cm, 4.0 cm, and 5.0 cm long strips of skin from the thoracic and abdominal areas, and 1.0 cm, 2.0 cm, and 3.0 cm long strips on the distal lateral portion of the hind leg, generally along the skin tension lines (*Irwin, 1966*). These were delineated with a permanent marker and the epidermis, dermis, and subcutis excised with a scalpel. All samples were approximately 1 cm wide. After removal, the samples were again measured using calipers (post-removal measurement), and then labeled and placed in 10% neutral buffered formalin. After 24–26 h of fixation, all samples were laid out, straightened, and measured (post-fixation measurement).

The response variable was the size of the sample. The factors that could affect size were species (dog/cat), gender (male/female), location (abdomen, thorax, leg) and time (initial, post-removal, post-fixation). The design was a split plot analysis of variance with the grouping factors of species and gender and repeat factors of location and time. Preliminary analysis (SAS PROC MIXED) yielded a significant effect of species, location, and time, with gender and all interactions associated with $p$-value $>0.34$. However, the errors were not normally distributed (determined via normal probability plot and the Shapiro–Wilk test). Therefore, final analysis was by means of nonparametric analysis, with the response variable being the size of the sample averaged over the five initial measurements (for example, for thorax and abdomen measurements, the initial measurement was the average of 1, 2, 3, 4, and 5 cm, giving 3 cm). Wilcoxon rank sum test was used to compare gender and species, with size of sample averaged across the other factors, and Wilcoxon signed rank test used to compare locations and time (for the dog and cat separately).

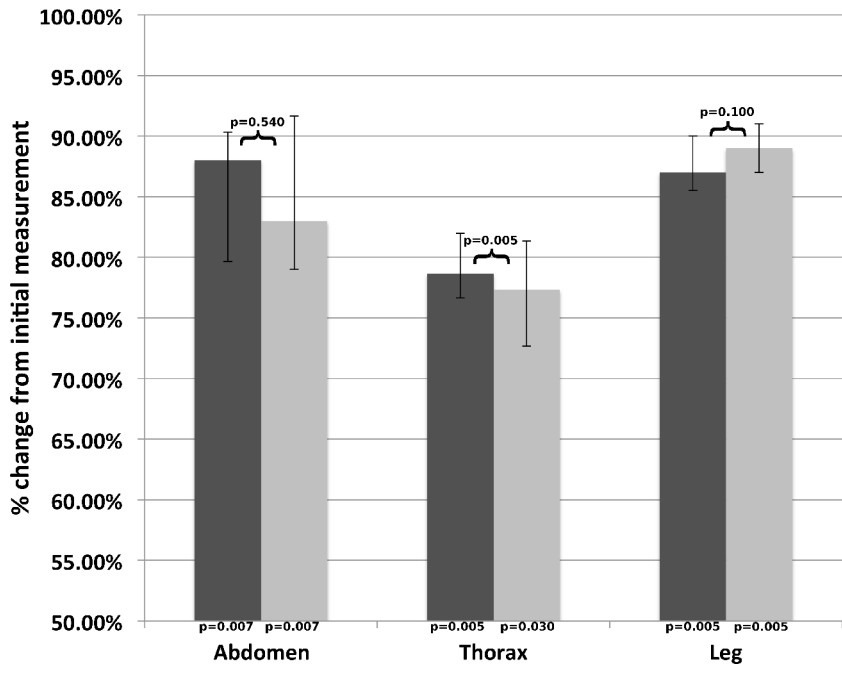

**Figure 1** **Aggregate median length (dog).** Median measurements for each site as a percentage of the initial measurement; post-removal in dark grey, and post-fixation in light grey. Error bars represent the 25th and 75th quartiles. *p*-values below each column are for comparisons with the initial measurement; *p*-values above the columns are for comparisons between both values.

Data were reported as the median (25th and 75th interquartile range). For comparisons of gender and species, a *p*-value <0.05 was considered significant. For multiple comparisons of location and time, a *p*-value <0.017 (0.05/3) was considered significant.

## RESULTS

The biggest difference in the size of the skin samples in dogs occurred immediately after removal (Fig. 1). The changes between the post-removal and post-fixation measurements varied, with some showing shrinkage and others expansion. The post-removal measurements were statistically different from the initial measurement for all measurements, but not from the post-fixation measurements aside from the thorax, which shrunk. There was no statistically significant difference between males and females.

Overall, the largest difference in size of the samples in cats occurred immediately after removal (Fig. 2) and, except for the 1.0 cm measurements and the 4.0 cm measurement from the abdomen, was the result of shrinkage. The post-removal measurements were statistically different from the initial measurement for the leg, but not the thoracic or abdominal samples, or for any post-fixation measurements. There was no statistically significant difference between males and females.

There were no differences between genders when measurements were averaged across all locations, times and species ($p = 0.45$). There was a significant difference in skin shrinkage in both dogs and cats between the thorax and abdomen compared to the leg across all time points ($p = 0.005$ for both species and locations). In addition, there was a

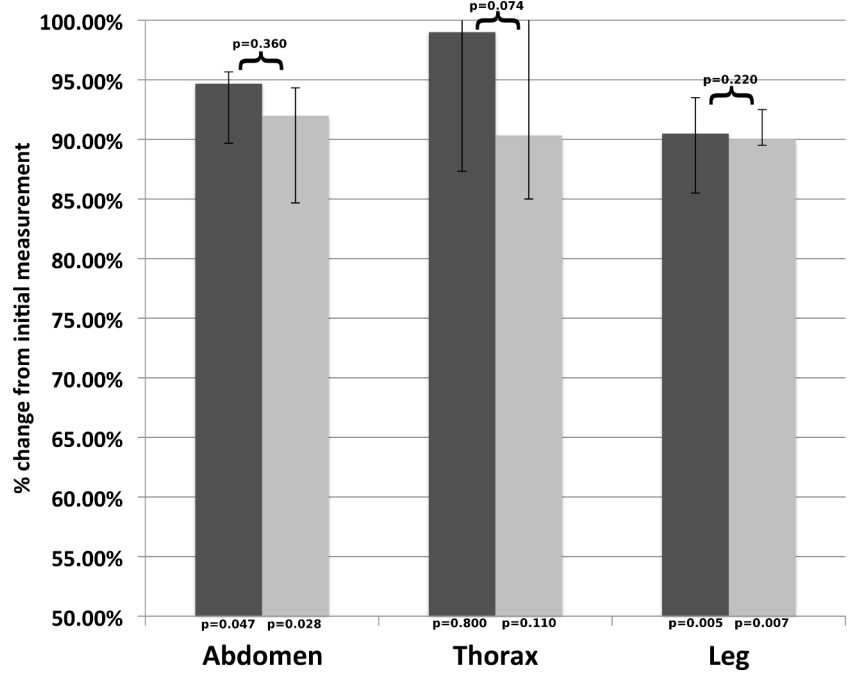

**Figure 2 Aggregate median length (cat).** Median measurements for each site as a percentage of the initial measurement; post-removal in dark grey, and post-fixation in light grey. Error bars represent the 25th and 75th quartiles. *p*-values below each column are for comparisons with the initial measurement; *p*-values above the columns are for comparisons between both values.

significant difference in skin shrinkage between the abdomen and thorax locations after removal in dogs ($p = 0.017$). Measurements from dogs were different than those from cats ($p = 0.004$). The remainder of differences between thorax and abdomen locations for dogs and all differences between the abdomen and thorax in cats were not significant.

## DISCUSSION

The statistical analysis suggests that differences in length of skin samples removed appear to be due to skin elasticity rather than effects of formalin. Dehydration or tissue loss are unlikely to play a role given the short time frame (less than a minute) between tissue excision and post-removal measurement. No post-fixation measurements were significantly different than the post-removal measurements, while all the post-removal measurements from the dog samples and the leg measurements from the cat were significantly different than the initial measurement. In fact, while there was no statistical significance, some samples expanded, rather than shrank, in formalin.

Few studies to date have been done comparing skin elastin levels between different species. While elastin is a relatively minor component of the skin's extracellular matrix, it plays a primary role in skin elasticity (*Tzaphlidou, 2004*) and tends to become damaged by solar exposure. However, elastin levels have been found to decrease with age (*McGavack & Kao, 1963*). While all animals appeared similar in age, we had no specific history for any of the animals in our tests. Therefore, some differences may be due to age variation or solar

exposure. In addition, the infiltration of the skin by neoplasia will inevitably replace elastin fibers with neoplastic cells, which may also alter the effects of surgical excision on the skin.

Interestingly, several previous studies examining the effects of formalin on tissues have found conflicting results. *Reimer et al. (2005)* used oval sections of skin taken antemortem to examine the same question, and found an overall decrease in combined length and width, and an increase in thickness. However, this study only examined one sample from each site per dog, and all samples from a particular site were the same size. *Blasdale et al. (2010)* examined this in basal cell carcinomas excised from people, and found skin shrinkage of 11–19% after formalin fixation. However, *Dauendorffer et al. (2009)* examined 82 excised human skin samples, and found that the most significant shrinkage occurred after removal and prior to formalin fixation. *Kerns et al. (2008)* examined 97 samples, and found similar shrinkage after excision rather than after formalin fixation.

The reason for these differences is unclear. The samples in this study are generally along the described skin tension lines for the dog. As the study by *Reimer et al. (2005)* examined both length and width, differences may be due to shrinkage along versus against lines of tension. Antemortem skin samples may be more sensitive to formalin fixation rather than recent post-mortem samples; however, human-based studies examining antemortem samples have also found disparate results (*Blasdale et al., 2010*; *Dauendorffer et al., 2009*).

Finally, this study also highlights the need for discussions between surgeons and pathologists about margin measurements. When evaluating clinical cases, it is important to determine whether a study uses measurements on the animal, after samples are excised, or histopathologic measurements to determine completeness of excision. Further, future studies evaluating margins should determine both gross and histologic margins necessary for surgical cure to be useful to both pathologists and surgeons.

## CONCLUSIONS

While this study, similar to others, found no correlation with gender, the site does seem significant in cats, as samples from the leg were the only ones with significant shrinkage. A majority of previous studies have found similar results, as *Reimer et al. (2005)* and *Dauendorffer et al. (2009)* both found that limb lesions had more significant shrinkage than body lesions. It is possible that the total sample size may play a role in the amount of shrinkage; the two human studies to examine this (*Dauendorffer et al., 2009*; *Hudson-Peacock, Matthews & Lawrence, 1995*) have conflicting results. Finally, as this study only examined healthy, adult animals, it does not determine if health and age may play a role in tissue shrinkage.

While this experiment provides another data point in the examination of the effects of formalin on skin samples, it also confirms that the sample measurements taken on the animals prior to surgery must be altered to take skin elasticity into account, as studies examining surgical margins necessary for cure examine margins after tissue processing for histopathology. Consideration of skin tension lines, skin elasticity, and surgical site should all play a role in determining the margins needed to produce the final histologic sample margin. Pathologists must also ensure that the criteria used to evaluate sample margins

are consistent with published studies. Further studies examining the shrinkage of samples from a variety of neoplasms, animal species, and locations are needed to determine the effects of different neoplasms, ages, and species on sample margin determination.

## ACKNOWLEDGEMENTS

The authors would like to thank Dr. Joseph Hauptman for performing statistical analyses and Patrick Knisley for his technical assistance.

### Funding

Funding was provided via internal funds from the University of Florida College of Veterinary Medicine. The funders had no role in study design, data collection and analysis, decision to publish, or preparation of the manuscript.

### Grant Disclosures

The following grant information was disclosed by the authors:
University of Florida College of Veterinary Medicine Startup Funding.

### Competing Interests

The authors declare they have no competing interests.

### Author Contributions

- Jaimie L. Miller conceived and designed the experiments, performed the experiments, analyzed the data, wrote the paper, reviewed drafts of the paper.
- Michael J. Dark conceived and designed the experiments, performed the experiments, analyzed the data, contributed reagents/materials/analysis tools, wrote the paper, prepared figures and/or tables, reviewed drafts of the paper.

### Animal Ethics

The following information was supplied relating to ethical approvals (i.e., approving body and any reference numbers):

University of Florida Institutional Animal Care and Use Committee: Approval #200902876.

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
