# Peer review of "Evaluation of the effect of formalin fixation on skin specimens in dogs and cats"

_PeerJ, doi:10.7717/peerj.307_

## Round 0.1 · original submission · Major Revisions

Kindly address the issues raised by the reviewers.

·

Basic reporting

No Comments

Experimental design

No Comments

Validity of the findings

No Comments

Additional comments

Very useful study with thorough statistical analysis!

Reviewer 2 ·

Basic reporting

The following standard was not met: "The article should include sufficient introduction and background to demonstrate how the work fits into the broader field of knowledge. Relevant prior literature should be appropriately referenced."

Please consider the folllowing to meet this standard:
- It should be clearly stated in the Introduction that the tissue samples were take from animals post-mortem.
- Additional background/references regarding how sufficient surgical margins are generally recommended to be obtained should be included in the Introduction.
- A clear objective or study goal should be stated in the last portion of the Introduction.

The following standard was met, however improvement is suggested: "The structure of the submitted article should conform to one of the templates. Significant departures in structure should be made only if they significantly improve clarity or conform to a discipline-specific custom."
- I suggest separating the Results section from the Discussion section for easier read or adding a "Conclusions" section. Most of the Results and Discussion section presented is a report of the results and more needs to be included in the way of discussion.
- Consider relating the results back to the potential effect on the common neoplasms listed in the Introduction for a more robust discussion.

Experimental design

Comments on Methods:
-Was a statistical analysis conducted before animal numbers were determined? The number of samples takes seem to be significantly less than the previous papers referenced in the Results and Discussion section.
- The addition of histological analysis and photographs (if significant changes are seen) of skin samples would greatly strengthen this study/presentation of data.

Validity of the findings

The data is adequately reported, although further discussion could be had to describe the species differences in skin elastin and how these results could change in the face of neoplasia. A statement could be made regarding further studies to assess shrinkage under varying conditions.

The Discussion could be expanded upon with conclusions; how does the statistical analysis reveal that the differnece in length of skin samples is due to elastin? Perhaps additional information can be provided regarding skin anatomy and the role of elastin, referencing additional studies assessing skin structure and physiology.

Additional comments

I strongly encourage the authors to exapand on the Introduction and Discussion sections to present their findings more completely.

Reviewer 3 ·

Basic reporting

This is a well-written manuscript.
1. The fourth sentence of the abstract is not clear until after the manuscript is fully read, in that the difference between "after removal"and "after formalin fixation" are not clearly defined. It is suggested that the abstract be worded to briefly outline the experimental design, primarily focusing on the three time points that were evaluated. This will provide defined terminology for use in the results portion of the abstract. The terminology used within the figures should be used consistently throughout the manuscript, including the abstract and materials and methods to improve clarity.

2. When describing the collection of samples, it is suggested to include orientation to tension lines (which is not described until the results section) and to include thickness of samples that were collected (or tissue plane).

3. The y axes of the figures lack titles.

Experimental design

No comments.

Validity of the findings

1. Regarding the last paragraph of the Conclusion section, the last sentence (lines 120–122) refers to the Simpson 2004 reference. When reviewing that reference, this reviewer interprets Simpson et al as measuring the margins prior to removal (e.g., "initial measurements" to use the terminology in this manuscript). Additionally, this current manuscript does not consider the full effect of "tissue processing" (line 119–120), as paraffin embedding and slide processing were not evaluated in this submitted manuscript. Thus, the conclusion may be better worded as the pathologist should be careful to ensure that the established criteria being used are from manuscripts describing post-fixation margins.

2. It is recommended for the figures to leave out the 3.0 cm bars for the initial measurements, as these are the author's defined baseline and not measured data (thus the reason that there are no error bars). The figures should then be reformatted to show the percent, proportion, or value, etc. of the post-removal and post-fixation sample sizes.

3. This reviewer finds the data figure and tables a bit confusing. Is it correct that Figure 1 corresponds to the "3.0 cm" row in Table 1? If not, the manuscript should be re-worded to clarify the differences. If they are the same, then the bars do not seem to match the tabular data. For instance, the bar representing the median post-removal abdomen data appears to be at 2.65 cm, which is 88.33% of 3.0 cm (the initial measurement), which is an 11.67% reduction. However, the 3.0 cm x Abdomen cell of Table 1 reports a 24.37% reduction as compared to the initial measurement.

4. The tabular and graphical data indicate that there is shrinkage post removal; however, the tabular data indicates that in most cases this is completely reversed (negative values), but the figures show the post-fixation data as similar to the post-removal data. Is the tabular data indicating percent change to the initial measurement or percent change to the previous time point (so the post-fixation value is the percentage of the post-removal measurement)?

5. Are the P values correct in Figure 1? For instance, the reported P values indicate more statistical significance for the Thorax post-removal vs. post-fixation comparison in contrast to the initial measurement vs. post-fixation.

Additional comments

No comments.

---

## Round 0.2 · Minor Revisions

The second reviewer raises some important concerns on the data representation and inadvertent typos, kindly address these.

Reviewer 1 ·

Basic reporting

No comments.

Experimental design

No comments.

Validity of the findings

No comments.

Additional comments

This is a well written manuscript. The revisions have greatly improved the presentation of this data.

Reviewer 2 ·

Basic reporting

1. The abstract should clarify that the stated no shrinkage post-fixation is relative to post-sampling, not relative to on animal. This relates to several comments below as well.

2. Discussion/line 164: delete repeated word “specific”

Experimental design

No comments.

Validity of the findings

1. The tabular data and the graphs remain confusing. The tabular numbers for “post-fixation” appear to be relative to “post-removal” even though the table description indicates they are relative to “initial measurement” (“initial measurement” is interpreted by this reviewer to mean “on animal”).
1a. For instance, all three data points for the Leg/Post-fixation are negative, indicating expansion. When the data is then aggregated into Figure one (which is also “a percentage of the initial measurement”) it shows that the Leg/Post-fixation samples are expanded relative to the post-removal samples, but are still smaller than the “initial measurement/on animal”. Please explain how three negative median values (expansion) in table 1, leg column, do not result in a bar above 100% (expansion). Not even the quartile error bar indicates expansion compared to the original measurement in Figure 1 for the leg?
1b. It appears this is resulting from the editing of the results section, which previously had more focus on the post-removal vs. post-fixation comparison, whereas the new version focuses on the post-fixation vs. initial measurement comparison. Please verify the tabular numbers in both tables, and if they are correct relative to “initial measurement”, then this needs to be better explained in the text how the tabular data can look so different from the graphical data.
2. Results/first paragraph
2a. It is interpreted that the second to last sentence (lines 85-87) is indicating that there are no statistical differences between the post-removal and the post-fixation measurements (and this is restated again in lines 155-156 and in the abstract). The graphical representation of the data appears to confirm that (the dark and light gray bars are of similar height with similar quartile bars); however, the P-value listed above the thoracic sample columns (“p=0.005”) indicates a significant (< 0.017 is indicated in the methods as the cut-off) change. Is the thorax post-sample vs. thorax post-fixation statistically significant or not? If it is not, please explain in the methods the criteria used for significance. If the listed P value is correct (which was the author's first response to this query), then the text of the manuscript needs to be corrected to indicate that in at least one circumstance there was a significant difference between the post-sampling and the post-fixation samples.
3. Results/second paragraph:
3a. The first sentence appears to be describing the tabular data, but references the figure; likewise, the second sentence appears to describe the graphical data (there are no statistical demarcations in the table), but references the table. This should be corrected.
3b. Consider rewording the first sentence to the “Overall the largest difference in size of the samples in cats occurred immediately after removal and, except for all 1.0 cm measurements and the 4.0 cm measurement from the abdomen, were the result of shrinkage.” This reviewer interpreted the original sentence to indicate that the exceptions were based on absolute magnitude of change, not direction of change, and the 4.0 cm abdominal measurement is relatively small (but it is negative).
3c. Please note: the original first sentence of this paragraph referenced the 4.0 cm thorax sample as expanding, but the table indicates it is the 4.0 cm sample from the abdomen. Regardless of any changes made based on suggestion 3b, please correct either the text or table appropriately.

Additional comments

No comments.

---

## Round 0.3 · accepted · Accept

Congratulations! The comments from the reviewer appear to be adequately addressed.

There remain a few minor points that should be corrected in your manuscript prior to publication.
1. Abstract is missing
2. Figure 2 is not cited in the revised manuscript.
3. Figure legends are missing.
4. Page 2, line 43: 'humans' would be better substituted for 'people'
5. Page 3, line 69: please refer to the statistical analyses program/model and then software.